# DEPfold: RNA Secondary Structure Prediction as Dependency Parsing

**Ke Wang**    **Shay B. Cohen**
School of Informatics, The University of Edinburgh
{k.wang-72,scohen}@ed.ac.uk

## Abstract

RNA secondary structure prediction is critical for understanding RNA function but remains challenging due to complex structural elements like pseudoknots and limited training data. We introduce DEPfold, a novel deep learning approach that re-frames RNA secondary structure prediction as a dependency parsing problem. DEPfold presents three key innovations: (1) a biologically motivated transformation of RNA structures into labeled dependency trees, (2) a biaffine attention mechanism for joint prediction of base pairings and their types, and (3) an optimal tree decoding algorithm that enforces valid RNA structural constraints. Unlike traditional energy-based methods, DEPfold learns directly from annotated data and leverages pretrained language models to predict RNA structure. We evaluate DEPfold on both within-family and cross-family RNA datasets, demonstrating significant performance improvements over existing methods. DEPfold shows strong performance in cross-family generalization when trained on data augmented by traditional energy-based models, outperforming existing methods on the bpRNA-new dataset. This demonstrates DEPfold's ability to effectively learn structural information beyond what traditional methods capture. Our approach bridges natural language processing (NLP) with RNA biology, providing a computationally efficient and adaptable tool for advancing RNA structure prediction and analysis.[1]

## 1 Introduction

Ribonucleic acid (RNA) molecules play crucial roles in biological processes, including gene expression regulation, protein synthesis, and gene editing systems such as CRISPR-Cas9 (Atkins et al., 2011; Sullenger & Nair, 2016; Grabow & Jaeger, 2014). RNA consists of an ordered sequence of nucleotides, each containing one of four bases: *adenine (A)*, *guanine (G)*, *cytosine (C)*, and *uracil (U)*. This sequence is referred to as the RNA's *primary structure*. These bases can form pairs, defining the *secondary structure*, as illustrated in Figure 1.

The function of RNA is closely related to its secondary structure. Accurate prediction of these structures is essential for understanding RNA functional mechanisms, designing RNA-targeted drugs, and studying RNA evolution (Puzzarini & Barone, 2018; Saini et al., 2021). While experimental methods such as X-ray crystallography, nuclear magnetic resonance, and cryo-electron microscopy can determine RNA structures (Zhang et al., 2022; Kappel et al., 2020), these approaches are generally limited by low throughput and high costs. The RNAcentral

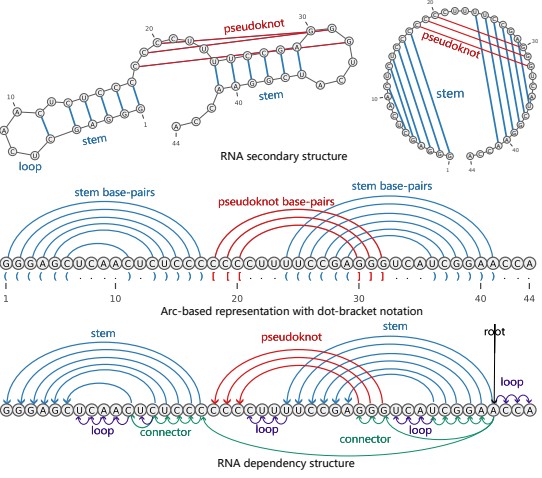

Figure 1: RNA secondary structure example and its corresponding dependency structure

---

[1]Our code is available at https://github.com/Vicky-0256/DEPfold.git.

database has cataloged over 24 million non-coding RNA sequences, yet only a tiny fraction have experimentally determined structures (The RNAcentral Consortium, 2016). RNA secondary structure prediction is a fundamental problem in biology, and serves wetlab biologists in tasks such as drug discovery (Oluoch et al., 2018) and cell analysis (Sethi et al., 2013).

Currently, there are two main methods for RNA structure prediction: energy-based methods and statistical and deep learning methods (Hajiaghayi et al., 2012). With energy based methods, the main prediction algorithm relies on minimizing, in an inference procedure, an energy objective that describes the alignment between the different nucleotides in the RNA sequence. Deep learning methods, on the other hand, rely on the usual machine learning setup, with a training set from which a structured prediction model is learned.

While energy-based models are overall more stable because they do not rely on training and are not sensitive to the distribution of the data they were trained on as deep learning models, their overall performance is often not very high when compared to deep learning models tested *within distribution*. Deep learning models do not generalize well out of their training distribution, because the amount of RNA sequenced data is small, for example, compared to proteins, where very large models such as AlphaFold (Jumper et al., 2021) are state of the art.

We balance these two issues, and motivated by NLP research on dependency parsing,[2] we develop **DEPfold**, an RNA structure prediction model, in which dependency parsing supports the following:

- A more detailed annotated structure than just basic alignment between nucleotides. This refined structures makes the learning problem easier for the underlying dependency model, as it provides additional constraints to the RNA structure, such as explicitly modeling *unaligned* nucleotides.
- A nested structure that follows the underlying assumptions for RNA structures. Pseudoknots[3] excluded, there is an underlying understanding that RNA structures are nested, and this is the reason why often models such as context-free grammars have been used for them (Knudsen & Hein, 1999; Do et al., 2006a). Our dependency model, while locally directly models the alignment between the different nucleotides, also has a global constraint in the form of a dependency trees that follows a nested structure.
- When pseudoknots are being included in the prediction problem, we can support that additional modification by changing the underlying inference algorithm for the dependency parsing model. Pseudoknots, we find, are equivalent to adding a certain level of *non-projectivity* to the dependency parsing model, as both notions indicate edges that cross each other in the dependency tree.
- Our dependency formulation model is built in such a way that it is easy to incorporate further constraints to the inference problem, such as energy-based constraints. This yields a hybrid model that balances well between learning from data and relying on a physical energy model originating in molecular biology (Zuker & Stiegler, 1981).
- Our approach relies on an encoding model to transform the RNA sequence into a sequence of embeddings. This component is plug-in, and can be used, for example, with a foundation model for RNA structures.

DEPfold adapts the biaffine parser of Dozat & Manning (2016), which uses deep biaffine attention to compute dependency relations. This parser decomposes the task into two modules: one predicting the existence of directed edges between nucleotides, and another predicting the best label for each potential edge. By transforming RNA structures into labeled dependency trees, DEPfold leverages the power of dependency parsing while maintaining biological constraints. This approach effectively predicts both the existence and types of base pairings, including pseudurknots, and can handle RNA sequences exceeding 3000 nucleotides. We conduct extensive experiments to compare DEPfold with state-of-the-art methods on several benchmark datasets, demonstrating its superior performance, particularly in predicting pseudoknots and long-range interactions. Moreover, DEPfold shows strong cross-family generalization when trained on augmented data, suggesting effective learning of structural information beyond traditional methods.

---

[2]Dependency parsing is the problem of identifying a syntactic structure for natural language sentence by identifying head-dependent relationships between words. See Kübler et al. (2009)

[3]Pseudoknots are a relatively unique type of bonds between nucleotides that violate nestedness in a tree.

## 2 BACKGROUND AND NOTATION

Our RNA structure prediction algorithm relies on **dependency parsing**, a method in NLP that provides syntactic structures to natural language utterances (Kübler et al., 2009). These structures originate in the syntactic theory of Tesnière (2015) called dependency grammar. At the core of this theory, it is stipulated that there is a head-dependent relationship between words in a sentence, described as edges between the different words that all together constitute a dependency parse tree. More formally, given an input sentence $x = w_0 w_1 \ldots w_n$, where $w_i \in \Sigma$ (an alphabet or a vocabulary), a dependency tree, as illustrated in Figure 1, is defined as $y = \{(i, j, \ell), 0 \leq i \leq n, 1 \leq j \leq n, \ell \in \mathcal{L}\}$, where $(i, j, \ell)$ represents a dependency relation $\ell \in \mathcal{L}$ from the head word $w_i$ to the modifier word $w_j$. In our case, the words are the nucleotides, and the edges between them describe alignment (or lack of) relationships between them. Dependency trees can be directed or undirected. While the alignments between nucleotides is undirected (corresponding to stem structures), we use a directed formalism to model more freely both alignment between nucleotides and lack thereof.

RNA consists of an ordered sequence of nucleotides, each containing one of four bases: *adenine (A)*, *guanine (G)*, *cytosine (C)*, and *uracil (U)*. Hence, we denote by $\mathcal{X} = \Sigma^*$ where $\Sigma = \{A, G, U, C\}$, the set of RNA sequences, by $\mathcal{Y}$ the set of RNA structures (in the form of Figure 1) and by $\mathcal{Z}$ the set of all dependency trees over RNA sequences from $\mathcal{X}$. In RNA secondary structures, each nucleotide pairs with at most one other nucleotide, forming structures such as stems and loops(Turner & Mathews, 2009). For clarity, we refer to these base-paired structures as *stems*, and the unpaired nucleotides form *loops*, as illustrated in Figure 1. A unique structural motif frequently observed in RNA is the *pseudoknot*, a double-helical structure formed by base pairing between single-stranded regions within stem-loop structures and external complementary sequences(Naderi et al., 2021). Formally, a pseudoknot occurs when an RNA has two base pairs, $i$–$j$ and $i'$–$j'$, such that $i < i' < j < j'$ (Achawanantakun & Sun, 2013). Pseudoknots can function as independent elements or as parts of complex RNA structures, participating in stabilization, replication, RNA processing, toxin inactivation, and gene expression control (Brierley et al., 2007; 2008; Giedroc & Cornish, 2009). Therefore, predicting pseudoknots is of significant importance.

In this study, we adopt two methods to represent RNA secondary structures. The first is the *arc representation*, where nucleotides are depicted as vertices and hydrogen bonds as arcs (Figure 1). For secondary structures without pseudoknots, all arcs are nested or parallel; crossed arcs indicate the presence of pseudoknots. The second method is the *dot-bracket notation*, where a `.` represents an unpaired base (loop base), and paired brackets denote base pairs. In this notation, base pairs forming stems are denoted by parentheses '()', and base pairs forming pseudoknots can be distinguished from stems by using square brackets'[]', angle brackets '$\langle\rangle$', and so on.

RNA secondary structures are crucial for determining the molecule's three-dimensional conformation and functional properties (Tinoco Jr & Bustamante, 1999). Accurate prediction of these structures aids in inferring tertiary interactions and has applications in drug design (Khatoon et al., 2014), gene regulation studies (Mortimer et al., 2014), and RNA-based therapeutics (Yin et al., 2014). By modeling stems, loops, and pseudoknots, we aim to capture the full complexity of RNA secondary structures, enabling more accurate predictions and a deeper understanding of RNA functionality.

## 3 DEPFOLD: METHODOLOGY

### 3.1 TRANSFORMATION OF RNA STRUCTURES TO DEPENDENCY

At the core of our algorithm, there is a mapping between RNA structures and dependency trees. Let $\tau \colon \mathcal{Y} \to \mathcal{Z}$ be such a mapping. The dependency tree $z = \tau(y)$ attached to $y$ includes all the alignment information in $y$, but also additional annotation, such as relationships of "lack of alignment." This additional fine-grained annotated structure further constrains and informs the learning algorithm, and also formulates the basic alignment relationship as a nested structure, following works for grammar-based and energy-based RNA structure prediction (Rivas & Eddy, 1999; Zuker & Stiegler, 1981; Dowell & Eddy, 2004).The transformation process involves four key steps (detailed in Appendix A. and B):

**Sequence Partitioning** Given an RNA sequence $x \in \mathcal{X}$ and its secondary structure $y \in \mathcal{Y}$ represented in bracket-dot notation, we partition $x$ into three subsets based on the structural information:

- $S$: The stem sequence, comprising nucleotides forming stem structures. In bracket-dot notation, these are represented by matching parentheses '(' and ')'.
- $P = \{P_1, \ldots, P_k\}$: A set of pseudoknot sequences, where each $P_i$ contains nucleotides forming a distinct pseudoknot. In extended bracket-dot notation, these are typically represented by different types of brackets (e.g., square brackets '[]', curly braces '{}', or angle brackets '$\langle\rangle$') to distinguish multiple pseudoknots.
- $L$: The loop sequence, containing all remaining unpaired nucleotides. These are represented by dots '.' in bracket-dot notation.

**Binary Tree Construction** For each sequence $Q$, where $Q$ can be either the stem sequence $S$ or any pseudoknot sequence $P_i$, we construct a binary tree $t_Q$ that preserves its nested structure. The construction follows these rules:

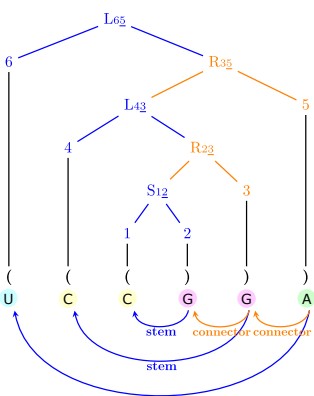

1. **Initialization**: Begin with the first complete pair of matching brackets in $Q$, denoted by positions $i$ and $j$, where $i < j$. This initial node is labeled as $S$ to represent *Start*.
2. **Directionality**: Assign the nucleotide at position $j$ as the *head* and the nucleotide at position $i$ as the *dependent*, forming an arc $(j, i)$ labeled appropriately (`stem` or `pseudoknot`).
3. **Traversal Order**: Proceed in a left-first, then right traversal to recursively construct left and right subtrees, maintaining the nested structures. Nodes are labeled with $L$ and $R$ to denote left and right subtrees, respectively.
4. **Arc Creation and Labeling**: For nucleotides under the same head node, we establish arcs to connect them in a right-to-left sequence. The arcs are labeled based on the structural relationship between the nucleotides(e.g. Arrow lines in Figure 1(bottom).):

Figure 2: Example of binary tree construction for an RNA stem sequence.

- **Stem Arcs**: If a pair of nucleotides is part of a stem structure, the connecting arc is labeled as `stem`.
- **Pseudoknot Arcs**: If a pair of nucleotides is part of a pseudoknot structure, the arc is labeled as `pseudoknot`.
- **Connector Arcs**: If the nucleotides are not part of either a stem or pseudoknot but are necessary to maintain tree connectivity, the arc is labeled as `connector`.

Figure 2 illustrates the process of constructing a binary tree for a stem sequence example, where numbers indicate the construction order. Here, $S$ represents the Start pair, and $L$ and $R$ denote the left and right subtrees, respectively. The pseudocode can be found in Appendix A, Algorithm 2.

**ROOT Construction** For RNA sequences that consist solely of stem structures, we select the overall head node of the stem tree $t_S$ as the head node of the entire sequence, denoted as node $g$. We then connect this node $g$ directly to the ROOT node, forming the complete dependency tree $z = \tau(y)$.

For RNA sequences that include pseudoknot structures, we first identify node $g$, and connect it to the ROOT node. We then integrate the stem tree $t_S$ with the pseudoknot trees $t_{P_i}$ to form a unified tree structure. The steps are as follows:

1. **Head Node Ordering**: Identify the head nodes of the stem tree and all pseudoknot trees, denoted as $H_S$ and $H_{P_i}$, respectively. Arrange these head nodes according to their positions in the original RNA sequence from right to left.
2. **ROOT Connection**: Designate the rightmost head node in the sequence as the overall head node of the entire sequence, node $g$. Connect this node $g$ to the ROOT node.
3. **Sequential Connection**: Connect the remaining head nodes to node $g$ using arcs labeled as `connector`, following the right-to-left direction. This forms a larger tree structure $t'$.

By connecting the stem and pseudoknot trees in this manner, we ensure that the combined tree accurately reflects the structural relationships and sequence ordering of the RNA molecule. The use of `connector` arcs preserves the necessary dependencies between different structural components.

**Tree Completion** To ensure that every nucleotide in the sequence is connected and the dependency tree is complete, we incorporate the loop nucleotides from $L$. For each nucleotide $x_k \in L$, we add

an arc $(\text{head}(k), k)$, where: $\text{head}(k) = \begin{cases} k+1, & \text{if } k < pos(g) \\ k-1, & \text{if } k > pos(g) \end{cases}$, where $pos(g)$ denotes the position index of a node $g$.

The arcs are labeled as `loop`, and this approach ensures that loop nucleotides are connected to their immediate neighbors, maintaining sequential order and tree completeness.

The final output of our transformation is the dependency tree $z \in \mathcal{Z}$, which encapsulates the complete RNA secondary structure. This dependency tree is composed of two integral components: (1) The set of arcs representing the structural connections between nucleotides, capturing both base-pairings and necessary connectors for tree completeness. (2) The set of labels assigned to each arc, indicating the type of relationship between nucleotides—such as `stem`, `pseudoknot`, `connector`, or `loop`. Every nucleotide in the RNA sequence is connected directly or indirectly to the `ROOT` node, ensuring that $z$ is a fully connected and rooted dependency tree. The final set of five labels we use is $\mathcal{L} = \{$ `loop`, `root`, `stem`, `connector`, `pseudoknot` $\}$.e.g. labels in Figure 1(bottom).

This comprehensive representation allows us to reformulate RNA secondary structure prediction as an optimization problem over dependency trees. Specifically, we aim to find the dependency tree $z$ that maximizes the conditional probability given the input RNA sequence $x$: $z^* = \text{argmax}_{z \in \mathcal{Z}} P(z \mid x)$, where $z^*$ is the predicted dependency tree, encompassing both the nucleotide connections and their labels.

## 3.2 UNDERLYING DEPENDENCY MODEL

Based on the transformation in §3.1, we use a biaffine parser as the core of our model for task parsing. Our proposed model, DEPfold, is composed of several key components. The overall architecture is illustrated in Figure 3.

**Foundation Model** We input an RNA sequence of length $N$, $x \in \mathcal{X}$, and embed it using a foundation model to obtain a contextual representation for each nucleotide in the sequence: $r_i = \text{Embed}(x_i)$. The foundation model can be a pretrained RNA foundation model, such as RNA-fm (Chen et al., 2022), or a general natural language foundation model like RoBERTa (Liu, 2019).[4] During this embedding phase, each nucleotide's representation is treated as a word, obtaining its corresponding token representation, and then deriving its contextual representation in the entire sequence.

**MLP Feature Extraction** Dozat & Manning (2016) first proposed using biaffine attention to compute dependency relations. This method formulates the dependency parsing task as labeling each edge in a directed graph, decomposing it into two modules: one predicting whether there is a directed edge between two words, and another predicting the best label for each potential edge. Similarly, we apply this approach to RNA secondary structure model prediction.

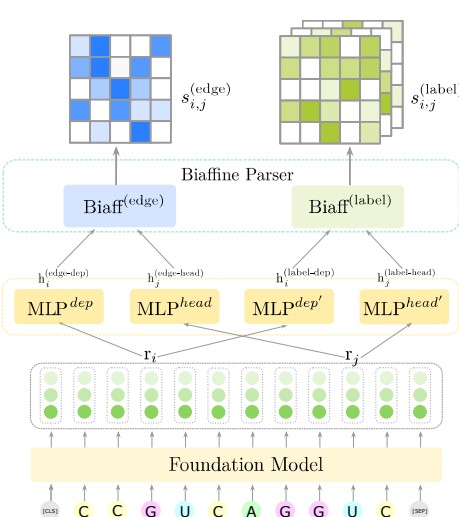

Figure 3: DEPfold Architecture.

For each nucleotide, we use MLPs to obtain dependent and head representations, as in the formulae:

$$h_i^{\text{dep}} = \text{MLP}^{\text{dep}}(r_i); \quad h_i^{\text{head}} = \text{MLP}^{\text{head}}(r_i).$$

where $h_i^{\text{dep}}$ and $h_i^{\text{head}}$ are the representation vectors of $x_i$ as a dependent nucleotide and a head nucleotide, respectively.

---

[4]While it may sound counter-intuitive to use a standard language model for RNA sequences, we found out RoBERTa can be applied to RNA sequences effectively. We believe it is because the model is fine-tuned on the RoBERTa's representations, which carry some information.

**Biaffine Scorer** We then use biaffine classifiers to predict scores for edges and labels separately. The biaffine classifier is a generalization of linear classifiers, including multiplicative interactions between two vectors:

$$\text{Biaff}(\mathbf{e}_1, \mathbf{e}_2) = \mathbf{e}_1^\top \mathbf{U}\mathbf{e}_2 + \mathbf{W}(\mathbf{e}_1 \oplus \mathbf{e}_2) + \mathbf{b},$$

$$s^{(\text{edge})}(i, j) = \text{Biaff}^{(\text{edge})}\left(h_i^{(\text{edge-dep})}, h_j^{(\text{edge-head})}\right); s^{(\text{label})}(i, j) = \text{Biaff}^{(\text{label})}\left(h_i^{(\text{label-dep})}, h_j^{(\text{label-head})}\right).$$

In the above, $s^{(\text{label})}(i, j)$ is a vector of length $|\mathcal{L}|$ ranging over labels. We denote by $s^{(\text{label})}(i, j, \ell)$ the specific value for a label $\ell \in \mathcal{C}$. The tensor $\mathbf{U}$ can be chosen as a diagonal matrix (making $u_{i,k,j} = 0$ when $i \neq j$) to save parameters. For the labeled parser, $\mathbf{U}$ will be $(d \times c \times d)$-dimensional, where $c$ is the number of labels. This operation is highly computationally efficient on GPUs.

**Training Loss** The model training is based on simple head nucleotide selection, without considering tree structure, and the losses for all words are accumulated in a mini-batch. For a standard head dependent pair $(x_i, x_j)$ with label $\ell^*$ in the training instance, the cross-entropy loss is:

$$L^{(\text{edge})}(i, j) = -\log \frac{\exp\left(s^{(\text{edge})}(i, j)\right)}{\sum_{0 \leq r \leq n} \exp\left(s^{(\text{edge})}(r, j)\right)}; \quad L^{(\text{label})}(i, j, \ell^*) = -\log \frac{\exp\left(s^{(\text{label})}(i, j, \ell^*)\right)}{\sum_{\ell \in \mathcal{L}} \exp\left(s^{(\text{label})}(i, j, \ell)\right)}.$$

In the overall system training, our goal is to minimize the sum of both losses over all dependency structures in the training data $\mathcal{D}$: $\sum_{z \in \mathcal{D}} \sum_{(i \rightarrow j, \ell^*) \in z} L^{(\text{edge})}(i, j) + L^{(\text{label})}(i, j, \ell^*)$.

**Decoding** After obtaining scores for all dependency relations, for the head selection of each nucleotide, we use the first-order Eisner algorithm (Eisner 2000; for projective structures) or graph spanning tree algorithms (Koo et al. 2007; for non-projective structures) to find the optimal tree $z^*$ without labels: $z^* = \arg\max_z \left[\sum_{i \rightarrow j \in z} s(i, j)\right]$. Based on the obtained optimal tree structure, we select the corresponding maximum label for each pair as the predicted value, leading eventually to the secondary structure $y^*$.

**Post-processing** RNA secondary structure only focuses on the connections we predict as stem and pseudoknots, and must comply with the rule that each nucleotide can only connect with one nucleotide, we first select those connections labeled as stem and pseudoknots from the predicted RNA dependency parsing structure and convert them into a contact map $M$.

We then use Softmax on $M$ to obtain probability distributions in the row and column directions:

$$\mathcal{C}(M) = \frac{\exp(M_{ij})}{\sum_{k=1}^{L} \exp(M_{kj})}; \quad \mathcal{R}(M) = \frac{\exp(M_{ij})}{\sum_{k=1}^{L} \exp(M_{ik})}.$$

Finally, we retain the maximum value in the row direction as the connection to obtain the final optimized contact matrix $M^*$, which ensures that the resulting connection structure is not conflicting: $M^* = \arg\max(\mathcal{R}(M))$.

## 4 EXPERIMENTAL SETUP

**Dataset** We evaluate DEPfold on four widely-used RNA structure prediction benchmark datasets:

RNAStrAlign (Tan et al., 2017) contains 37,149 structures from 8 RNA families. Following E2Efold (Chen et al., 2020) and MXfold2 (Sato et al., 2021), we processed the dataset, retaining 30,451 non-redundant structures. ArchiveII (Sloma & Mathews, 2016), comprising 3,975 structures from 10 RNA families, serves as a standard benchmark for classical RNA folding methods. Both datasets include sequences with pseudoknots.

bpRNA-1m (Singh et al., 2019) includes 102,318 structures from 2,588 RNA families. We processed it following SPOT-RNA (Singh et al., 2019), using CD-HIT (Fu et al., 2012) for redundancy removal and dataset splitting. bpRNA-new (Kalvari et al., 2017), derived from Rfam 14.2, contains sequences from 1,500 novel RNA families and is used to assess cross-family generalization. Consistent with UFold (Fu et al., 2022), we augmented the training data by randomly mutating bpRNA-new sequences and generating structure predictions using RNAfold (Lorenz et al., 2011b). Sequences in bpRNA-1m and bpRNA-new are shorter than 500 nucleotides and lack pseudoknots, contrasting with the first two datasets. Table 1 summarizes the key characteristics of each dataset.

**Baseline Methods** We compare our proposed DEPfold with several baseline methods, including:

Energy-based folding methods: CONTRAfold (Do et al., 2006b), RNAfold (Lorenz et al., 2011b), LinearFold (Huang et al., 2019)(using the thermodynamic free energy model from Vienna RNAfold(Lorenz et al., 2011b)), and RNAstructure (Reuter & Mathews, 2010)(using the ProbKnot(Bellaousov & Mathews, 2010) algorithm when predicting RNA datasets with pseudoknots). Learning-based folding methods: E2Efold (Chen et al., 2020), MXfold2 (Sato et al., 2021), UFold (Fu et al., 2022), and RFold (Tan et al., 2024).

**Evaluation Metrics** As standard in RNA structure prediction, we evaluated performance using precision (or PPV, Positive Predictive Value), recall (or SEN, sensitivity), and $F_1$ score. More specifically, the set of all aligned pairs are extracted from both the predicted structure and the reference structure, and the macro-averaged statistics are calculated based on them.

Table 1: Summary of datasets used in our experiments.

| Dataset | Subset | #Seq. | Len. Range |
|---|---|---|---|
| RNAStrAlign | Train | 28,969 | 30–1581 |
| | Val | 3,629 | 36–1693 |
| | Test | 2,810 | 57–1672 |
| ArchiveII | - | 3,975 | 28–2968 |
| bpRNA-1m | TR0 | 10,814 | 33–498 |
| | VL0 | 1,300 | 33–497 |
| | TS0 | 1,305 | 22–499 |
| bpRNA-aug | TR1 | 20,431 | 33–498 |
| bpRNA-new | - | 5,401 | 33–489 |

## 5 RESULTS

To evaluate the performance of our DEPfold model relative to baseline models, we conducted two sets of experiments, following a similar approach to that of Fu et al. (2022); Sato et al. (2021): (a) training the model on the RNAStrAlign training set and testing on the RNAStrAlign test set and the ArchiveII dataset; (b) training the same model on the bpRNA-1m training set (TR0) and testing on the bpRNA-1m test set (TS0), as well as training on bpRNA-1m augmentation set (TR1) and testing on the bpRNA-new (bpnew) dataset. For detailed information on the model training process, see Appendix C.

### 5.1 PERFORMANCE ON RNASTRALIGN AND ARCHIVEII

We assessed the performance of DEPfold on the RNAStrAlign test set, and the results are presented in Table 2. DEPfold achieved the best performance across all evaluation metrics. Compared to traditional energy-based algorithms, DEPfold demonstrated significant advantages over both traditional methods such as LinearFold and RNAfold and other deep learning methods such as E2Efold and Ufold. In addition, DEPfold surpassed other deep learning methods in both precision and recall. Finally, DEPfold can handle RNA sequences of arbitrary lengths, whereas E2Efold, RFold and UFold are constrained by a fixed maximum sequence length (e.g., 1,800 nucleotides).

Table 2: Performance comparison on RNAStrAlign test set.

| Method | Precision | Recall | $F_1$ |
|---|---|---|---|
| **DEPfold** | **0.988** | **0.983** | **0.985** |
| UFold | 0.959 | 0.965 | 0.962 |
| RFold | 0.942 | 0.780 | 0.797 |
| E2Efold | 0.866 | 0.788 | 0.821 |
| ContraFold | 0.600 | 0.647 | 0.621 |
| LinearFold | 0.523 | 0.576 | 0.547 |
| RNAfold | 0.515 | 0.568 | 0.539 |
| RNAstructure | 0.537 | 0.569 | 0.551 |

**Pseudoknot Prediction** To evaluate DEPfold's ability to handle complex RNA structures, particularly sequences containing pseudoknots, we analyzed sequences with pseudoknots in the RNAStrAlign test set. Following the methodology of E2Efold (Chen et al., 2020), we calculated the average $F_1$ score (Set $F_1$) for these sequences and counted the number of sequences where the presence or absence of pseudoknots was correctly predicted. Table 3 summarizes the evaluation results.

Table 3: Evaluation of pseudoknot prediction.

| Method | Set $F_1$ | TP | FP | TN | FN |
|---|---|---|---|---|---|
| **DEPfold** | **0.981** | 1258 | 20 | 1531 | 1 |
| Ufold | 0.838 | 159 | 111 | 1397 | 0 |
| E2Efold | 0.710 | 1312 | 242 | 1271 | 0 |
| RNAstructure | 0.474 | 975 | 306 | 1245 | 284 |
| Rfold | 0.447 | 685 | 55 | 1470 | 122 |

DEPfold achieved the highest $F_1$ score of 0.981 on this set, significantly outperforming other methods. It demonstrated high accuracy in identifying both pseudoknotted and non-pseudoknotted sequences, while maintaining low false positive and false negative rates. These results indicate that DEPfold exhibits exceptional capability in handling com-

plex RNA structures containing pseudoknots, showcasing its potential to effectively capture the intricacies of RNA structural complexity.

**Generalization to ArchiveII** To assess DEPfold's generalization ability, we directly tested the model trained on the RNAStrAlign training set on the ArchiveII dataset. Table 4 compares the performance of DEPfold with other methods on the ArchiveII dataset. DEPfold achieved an $F_1$ score of 0.920 on this dataset, significantly higher than other methods, and also obtained the highest precision and recall. This demonstrates that DEPfold not only achieves strong results on specific tasks but also has the capacity to adapt to a wide range of RNA secondary structure prediction tasks.

Table 4: Performance comparison on ArchiveII set.

| Method | Precision | Recall | $F_1$ |
|---|---|---|---|
| **DEPfold** | **0.941** | **0.907** | **0.920** |
| RFold | 0.931 | 0.899 | 0.911 |
| UFold | 0.876 | 0.890 | 0.881 |
| MXfold2 | 0.825 | 0.780 | 0.796 |
| E2Efold | 0.734 | 0.660 | 0.686 |
| ContraFold | 0.593 | 0.649 | 0.617 |
| LinearFold | 0.549 | 0.609 | 0.575 |
| RNAfold | 0.550 | 0.611 | 0.577 |
| RNAstructure | 0.554 | 0.601 | 0.573 |

Table 5: Performance comparison on bpRNA-TS0 set.

| Method | Precision | Recall | $F_1$ |
|---|---|---|---|
| **DEPfold** | **0.730** | 0.656 | **0.676** |
| RFold | 0.676 | 0.625 | 0.633 |
| UFold | 0.587 | **0.711** | 0.630 |
| MXfold2 | 0.519 | 0.646 | 0.558 |
| ContraFold | 0.482 | 0.655 | 0.541 |
| LinearFold | 0.447 | 0.633 | 0.510 |
| RNAfold | 0.446 | 0.631 | 0.508 |
| RNAstructure | 0.446 | 0.622 | 0.505 |

## 5.2 PERFORMANCE ON THE BPRNA DATASET

Following prior studies (Sato et al., 2021; Fu et al., 2022), we trained on bpRNA-TR0 and evaluated on bpRNA-TS0 using the best model obtained from bpRNA-VL0. Detailed results are presented in Table 5. DEPfold significantly outperformed UFold in terms of $F_1$ score, with an improvement of 7 percentage points, and achieved a 25% enhancement over traditional methods.

Table 6: Performance on long-range base pair prediction (bpRNA-TS0)

| Method | Precision | Recall | $F_1$ |
|---|---|---|---|
| **DEPfold** | **0.715** | 0.689 | **0.689** |
| UFold | 0.630 | **0.711** | 0.653 |
| RFold | 0.666 | 0.647 | 0.643 |
| ContraFold | 0.505 | 0.614 | 0.536 |
| LinearFold | 0.502 | 0.629 | 0.542 |
| RNAfold | 0.499 | 0.627 | 0.539 |
| RNAstructure | 0.490 | 0.603 | 0.523 |

**Long-range Interaction Prediction** To evaluate whether DEPfold contributes to improved prediction of long-range interactions, we adopted the same experimental approach as UFold. We used the TS0 dataset as the test set because it contains diverse sequences of varying lengths from various RNA families. For a sequence of length L, we define base pairs with a separation greater than L/2 as long-range base pairs. As shown in Table 6, DEPfold is able to predict long-range base pairs, outperforming UFold. Notably, DEPfold's performance in predicting long-range base pairs was comparable to, or even slightly better than, its performance in predicting short-range base pairs.

**Cross-family Evaluation on bpRNA-new** The bpRNA-new dataset is a cross-family benchmark dataset that poses significant challenges to purely deep learning methods because these families are not represented in the training set. To address this issue, MXfold2 proposed combining free energy minimization with deep learning methods, achieving performance similar to the thermodynamics-based ContraFold. The purely deep learning method DEPfold, which is based on data augmentation (TR1 dataset for training) from RNAfold, achieved an $F_1$ score of 0.621 on the bpRNA-new dataset, exceeding RNAfold by 0.65 percentage points. This result indicates that DEPfold can learn

Table 7: Performance comparison on bpRNA-new.

| Method | Precision | Recall | $F_1$ |
|---|---|---|---|
| **DEPfold** | **0.650** | 0.624 | 0.621 |
| UFold | 0.527 | 0.695 | 0.590 |
| RFold | 0.528 | 0.306 | 0.369 |
| MXfold2 | 0.585 | 0.710 | 0.632 |
| ContraFold | 0.578 | **0.736** | **0.639** |
| LinearFold | 0.551 | 0.719 | 0.615 |
| RNAfold | 0.552 | 0.720 | 0.617 |
| RNAstructure | 0.542 | 0.703 | 0.604 |

more useful structural information on top of what traditional methods capture. In contrast, UFold, which also uses data augmentation, performed slightly worse than RNAfold on this dataset.

## 5.3 ABLATION STUDY

To gain deeper insight into the contributions of various components of DEPfold, we conducted a series of ablation experiments.

**Pretrained Models** We compared the effectiveness of using RoBERTa and RNAfm as RNA sequence representation generators. which are denoted as DEPfold-Ro and DEPfold-fm, respectively. RoBERTa is a general-purpose large-scale language model, while RNAfm is a pretrained model specifically designed for RNA sequences. Table 8 presents the results. The DEPfold model based on RNAfm achieved an $F_1$ score of 0.618 on TS0, slightly lower than the version using RoBERTa. This suggests that although RNAfm is specialized for RNA sequences, the general language representations provided by RoBERTa perform better in this task.

Table 8: Ablation Study on Pretrained Models and Fine-Tuning (bpRNA-TS0).

| Method | Precision | Recall | $F_1$ |
|--------|-----------|--------|-------|
| DEPfold-Ro,FT | **0.730** | **0.656** | **0.676** |
| DEPfold-fm,FT | 0.719 | 0.574 | 0.618 |
| DEPfold-Ro,FrZ | 0.428 | 0.174 | 0.217 |
| DEPfold-fm,FrZ | 0.644 | 0.422 | 0.470 |

**Fine-Tuning vs. Freezing Pretrained Models** We further investigated the impact of fine-tuning versus freezing the pretrained models on DEPfold's performance. As shown in Table 8, fine-tuning the pretrained models (FT) significantly enhances the model's ability to predict RNA secondary structures. In contrast, freezing the pretrained models (FrZ) leads to substantial performance degradation, with the model nearly failing to learn meaningful representations for RNA secondary structure prediction. Interestingly, under frozen conditions, the model using RNAfm performed slightly better than the one with RoBERTa, possibly because RNAfm, being specifically pretrained on RNA sequences, had already captured some RNA-specific features during its pretraining phase.

**Optimal Tree Decoding** We evaluated the impact of the optimal tree decoding algorithm on DEPfold's performance by comparing the model with and without this strategy. The results are presented in Table 9, where "DEPfold w/o OT" denotes the DEPfold model without the Optimal Tree decoding strategy. The findings show that incorporating Optimal Tree decoding consistently enhances performance across both datasets. On TS0, the $F_1$ score increases from 0.671 to 0.676 when optimal tree decoding is used, primarily due to an increase in precision from 0.676 to 0.730. Similarly, on bpRNA-new, the $F_1$ score improves from 0.607 to 0.621. These enhancements demonstrate that optimal tree decoding effectively enforces valid RNA structural constraints during decoding, leading to more accurate and biologically plausible predictions.

Table 9: Ablation Study on Optimal Tree Decoding (bpRNA).

| Dataset | Method | Precision | Recall | $F_1$ |
|---------|--------|-----------|--------|-------|
| TS0 | DEPfold | 0.730 | 0.656 | 0.676 |
| | DEPfold w/o OT | 0.676 | 0.686 | 0.671 |
| bpRNA-new | DEPfold | 0.650 | 0.624 | 0.621 |
| | DEPfold w/o OT | 0.632 | 0.613 | 0.607 |

## 6 RELATED WORK

**Traditional Methods** Traditional approaches include alignment-based and single-sequence prediction methods. Alignment-based methods like the Sankoff algorithm (Sankoff, 1985) and its variants—Dynalign (Mathews & Turner, 2002) and Carnac (Touzet & Perriquet, 2004)—predict structures by identifying conserved motifs among homologous sequences. However, their effectiveness is limited by the limited number of RNA families in databases like Rfam (Kalvari et al., 2021) and low sequence conservation. Single-sequence methods, such as Vienna RNAfold (Lorenz et al., 2011a) and Mfold (Zuker, 2003), predict structures by minimizing free energy based on thermodynamic models. While effective for short sequences, they struggle with efficiency and accuracy on complex structures, particularly pseudoknots, which pose NP-complete challenges (Lyngsø & Pedersen, 2000). Tools like RNAstructure (Reuter & Mathews, 2010) extend capabilities to include pseudoknot prediction and external constraints but face computational bottlenecks with long sequences.

**Machine Learning and Deep Learning Methods** Deep learning models have been introduced for RNA structure prediction. SPOT-RNA (Singh et al., 2019) combines ResNet (Koonce & Koonce, 2021) and bidirectional LSTM (Hochreiter, 1997) but lacks constraints to ensure valid structures, affecting generalization (Amos & Kolter, 2017). E2Efold (Chen et al., 2020) uses convex optimization and algorithm unrolling to constrain outputs. UFold (Fu et al., 2022) uses convolutional networks on image representations of sequences, enhancing pseudoknot prediction. MXfold2 (Sato et al., 2021) combines deep learning with thermodynamic models, generalizing well across families. However, these models generally struggle with long sequences; for example, SPOT-RNA supports up to 2000 nucleotides, E2Efold up to 1800, UFold up to 600, and MXfold2 up to 1000 nucleotides.

**NLP-Inspired Methods** NLP techniques have been applied to RNA structure prediction. Linear-Fold (Huang et al., 2019) reduces prediction time from quadratic to linear using beam pruning (Huang et al., 2012), a popular heuristic widely used in computational linguistics, but cannot predict pseudoknots. ContextFold (Zakov et al., 2011) and CONTRAfold (Do et al., 2006b) use context-free grammars for base-pair dependencies, improving accuracy but still struggling with complex structures like pseudoknots. Matsui et al. proposed a new structural alignment algorithm based on pair stochastic tree adjoining grammars (PSTAGs) to align and predict RNA secondary structures, including pseudoknots. NLP methods improve computational efficiency and handle some complex structures, but integrating deep learning for RNA secondary structure prediction, especially for pseudoknots and long sequences, remains challenging.

## 7 CONCLUSION

In this work, we introduced DEPfold, a novel RNA secondary structure prediction framework that reformulates the task as dependency parsing by leveraging advanced NLP techniques. DEPfold transforms RNA structures into labeled dependency trees, uses a biaffine attention mechanism for accurate base pairing prediction, and uses optimal tree decoding to ensure biologically valid structures. Our approach significantly outperforms traditional energy-based methods and state-of-the-art deep learning models, achieving an F1 score of 0.985 on the RNAStrAlign dataset (which we note, though, is a within-family dataset) and effectively capturing complex features such as pseudoknots and long-range interactions. Furthermore, DEPfold demonstrates robust cross-family generalization on the bpRNA-new dataset, identifying structural nuances that conventional models miss. Ablation studies highlight the unexpected efficacy of general-purpose models like RoBERTa in fine-tuning, indicating potential for specialized pretraining strategies in RNA biology. While DEPfold marks a significant advancement, future work will address challenges in enhancing cross-family generalization and model interpretability by expanding training data diversity, optimizing computational efficiency, and improving interpretability to broaden DEPfold's applicability in diverse biological research contexts.

## LIMITATIONS

Our experiments on within-family datasets, such as RNAStrAlign and bpRNA-TS0, show that DEPfold achieves high $F_1$ score, indicating effective learning from the training data. However, when evaluated on cross-family datasets like bpRNA-new—which includes RNA families not represented in the training set—DEPfold's performance shows a smaller margin of improvement (outperforming RNAfold by only 0.65 percentage points). This suggests that while DEPfold can capture structural information beyond traditional methods, its ability to generalize to entirely new RNA families is still limited. See discussion by Szikszai et al. (2022) and Bernett et al. (2024).

Our data augmentation strategy, which uses RNAfold-predicted structures, while beneficial, may reinforce existing biases in traditional energy-based models. While effective in increasing training data, this approach may limit DEPfold's ability to learn non-traditional structural patterns. Future work should explore more diverse data augmentation techniques that introduce a wider range of structural variations. Additionally, scaling DEPfold to larger model sizes and datasets, and incorporating neural dependency parsing models like stack-pointer (Ma, 2018) and TreeCRF (Zhang et al., 2020a;b), could enhance its ability to learn from diverse syntactic representations and improve overall capabilities.

An interesting finding from our ablation studies is the superior performance of RoBERTa over RNA-fm in the fine-tuning stage. This result prompts us to rethink the optimal pretraining strategy for RNA structure prediction. RoBERTa's general language patterns seem to provide more valuable context for structure prediction than RNA-fm's RNA-specific features. This suggests that current RNA-specific models may not fully capture the characteristics needed for accurate structure prediction. Our findings suggest the need for developing more effective pretraining approaches specifically designed for RNA biology, which could potentially improve the accuracy and generalization capabilities of models like DEPfold.

ACKNOWLEDGMENTS

We would like to thank the anonymous reviewers, Grzegorz Kudla, Mengyu Wang and Ajitha Rajan for feedback and useful comments. We appreciate the provision of compute resources through the Baskerville cluster (University of Birmingham). This work was supported by the United Kingdom Research and Innovation (grant EP/S02431X/1), UKRI Centre for Doctoral Training in Biomedical AI at the University of Edinburgh, School of Informatics.

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

# A PSEUDOCODE FOR RNA SECONDARY STRUCTURE TO DEPENDENCY STRUCTURE

To clearly explain the algorithmic logic for converting RNA secondary structures into dependency structures, we present the following pseudocode. Algorithm 1 is the main program, which uses the get_pair function defined in Algorithm 2 to generate binary tree structures from stem and pseudoknot sequences. The Get_Pair function, during its processing, utilizes the Is_Connect function defined in Algorithm 3 for decision-making. When handling unpaired structures, the algorithm employs the get_pairs function defined in Algorithm 4.

The primary input to the algorithm consists of the RNA sequence, the base pairs of stems and pseudoknots, and a list of indices representing unpaired dots. Through this algorithm, one can derive the arc connections and labels of the dependency structure for the given RNA sequence.

---

**Algorithm 1:** Algorithm for Converting RNA Secondary Structure to Dependency Structure

**Input:** RNA sequence length $n$, stem base pairs $S$, pseudoknot base pairs $P$, unpaired base indices $D$, dot-bracket notation $dot$, connectivity table $ctList$

**Output:** Tree structure $arcs$ and $labels$('stem', 'pseudoknot', 'loop', 'connector', 'root')

1  Initialize empty lists: $last\_node\_list \leftarrow [\,], S_p \leftarrow [\,], P_p \leftarrow [\,], L_p \leftarrow [\,]$;
2  Initialize $dot\_indices \leftarrow \{i \mid dot[i] =' .'\}$;

   // Process Stem Pairs
3  **if** $S$ *is not empty* **then**
4      Initialize $dot \leftarrow ['.'] \times n$;
5      **foreach** $(l, r)$ *in* $S$ **do**
6        |  $dot[l-1] \leftarrow' ('$ , $dot[r-1] \leftarrow')'$;
7      **end**
8      Convert $dot$ to a string;
9      Identify $stem\_indices \leftarrow \{i \mid dot[i] \in \{'(','\,)'\}\}$;
10     $S_p \leftarrow$ get_pair$(dot[stem\_indices], stem\_indices)$;
11     Append $stem\_indices[-1]$ to $last\_node\_list$;
12 **end**

   // Process Pseudoknot Pairs
13 **if** $P$ *is not empty* **then**
14     **foreach** *set in* $P$ **do**
15       Initialize $dot \leftarrow ['.'] \times n$;
16       **foreach** $(l, r)$ *in set* **do**
17         |  $dot[l-1] \leftarrow' ('$ , $dot[r-1] \leftarrow')'$;
18       **end**
19       Convert $dot$ to a string;
20       Identify $pse\_indices \leftarrow \{i \mid dot[i] \in \{'(','\,)'\}\}$;
21       $have\_pair \leftarrow$ get_pair$(dot[pse\_indices], pse\_indices)$;
22       Extend $P_p$ with $have\_pair$;
23       Append $pse\_indices[-1]$ to $last\_node\_list$;
24     **end**
25 **end**

   // Combine Trees Together
26 Sort $last\_node\_list$ in decreasing order;
27 $connect\_pair \leftarrow \{(last\_node\_list[i], last\_node\_list[i+1]) \mid i = 0$ to $|last\_node\_list| - 2\}$;
28 Extend $S_p$ with $connect\_pair$;

   // Process Unpaired Bases
29 Initialize $result\_list \leftarrow [\,], temp\_list \leftarrow [\,]$;

---

**1** Continue the algorithm from previous part;
**2** **for** $i$ *from* 0 *to* $|dot\_indices| - 1$ **do**
**3**  | **if** $i = 0$ **or** $dot\_indices[i] = dot\_indices[i-1] + 1$ **then**
**4**  |  | Append $dot\_indices[i]$ to *temp_list*;
**5**  | **else**
**6**  |  | Append *temp_list* to *result_list*;
**7**  |  | $temp\_list \leftarrow [dot\_indices[i]]$;

**8** Append *temp_list* to *result_list*;
**9** $L_p \leftarrow$ get_pairs($result\_list$, $last\_node\_list[0]$);

   // Arc Creation and Labeling
**10** Adjust indices in $S_p$, $P_p$, $L_p$ to be based-1 (if necessary);
**11** Initialize arrays $arc \leftarrow [0] \times (n+1)$, $rel \leftarrow [0] \times (n+1)$;
**12** Initialize empty dictionaries $heads \leftarrow \{\}$, $relations \leftarrow \{\}$;

**13** **foreach** $(head, dep)$ *in* $S_p$ **do**
**14**  | $heads[dep] \leftarrow head$;
**15**  | **if** $(head, dep)$ **or** $(dep, head)$ *in* $S$ **then**
**16**  |  | $relations[dep] \leftarrow$ 'stem';
**17**  | **else**
**18**  |  | $relations[dep] \leftarrow$ 'connector';

**19** **foreach** $(head, dep)$ *in* $P_p$ **do**
**20**  | $heads[dep] \leftarrow head$;
**21**  | **if** $(head, dep)$ **or** $(dep, head)$ *in* $P$ **then**
**22**  |  | $relations[dep] \leftarrow$ 'pseudoknot';
**23**  | **else**
**24**  |  | $relations[dep] \leftarrow$ 'connector';

**25** **foreach** $(head, dep)$ *in* $L_p$ **do**
**26**  | $heads[dep] \leftarrow head$;
**27**  | $relations[dep] \leftarrow$ 'loop';

**28** **for** $i$ *from* 1 *to* $n$ **do**
**29**  | **if** $i$ *is in* heads **then**
**30**  |  | $arc[i] \leftarrow heads[i]$;
**31**  |  | $rel[i] \leftarrow relations[i]$;
**32**  | **else**
**33**  |  | $arc[i] \leftarrow 0$;
**34**  |  | $rel[i] \leftarrow$ 'root';

---

**Algorithm 2:** Function `get_pair` for binary tree construction in stem/pseudoknot Structures

---

**Input:** Sequence *seq* of symbols $'(', ')'$, and indices *idx*
**Output:** Paired indices *parse_pair* representing arcs

1   Initialize $i \leftarrow 0$, *parse_pair* $\leftarrow [\,]$;
2   **while** $i \neq -1$ **and** $|seq| > 1$ **do**
3      $index \leftarrow i$;
     // Determine `a`, `b`, `preindex`, `change_idx` based on `seq[index]`
4      **if** $seq[index] == 1$ **then**
5          $a \leftarrow seq[index - 1], b \leftarrow seq[index]$;
6          $preindex \leftarrow index - 1, change\_idx \leftarrow index$;
7      **else**
8          **if** $seq[index] == 0$ **and** $seq[index - 1] == 0$ **then**
9              $a \leftarrow seq[index - 1], b \leftarrow seq[index]$;
10              $preindex \leftarrow index - 1, change\_idx \leftarrow index$;
11          **else**
12              $a \leftarrow seq[index], b \leftarrow seq[index + 1]$;
13              $preindex \leftarrow index, change\_idx \leftarrow index + 1$;
14      $(c, d) \leftarrow$ `is_connect`$(a, b)$;
15      **if** $c \neq 2$ **then**
16          Append $(idx[change\_idx], idx[preindex])$ to *parse_pair*;
17          $seq[change\_idx] \leftarrow c$;
18          Remove $seq[change\_idx - 1]$ and $idx[change\_idx - 1]$;
19      $i \leftarrow i + d$;
20   **return** *parse_pair*

---

**Algorithm 3:** Function `is_connect` for Pairing Symbols

---

**Input:** Symbols $a$, $b$ (can be $'(', ')'$, 0, 1)
**Output:** New symbol $c$, index adjustment $d$

1   **if** $a ==' ('$ **then**
2      **if** $b ==' ('$ **then**
3          **return** $(2, 1)$ // Keep processing
4      **else**
5          **if** $b ==')'$ **then**
6              **return** $(0, 0)$ // Pair formed
7          **else**
8              **if** $b == 1$ **then**
9                  **return** $(0, -1)$ // Pair formed, move back

10   **else if** $a == 0$ **then**
11      **if** $b ==')'$ **then**
12          **return** $(1, 0)$ // Pair formed
13      **else**
14          **if** $b == 0$ **then**
15              **return** $(0, -1)$ // Pair formed, move back
16          **else**
17              **if** $b ==' ('$ **then**
18                  **return** $(2, 1)$ // Keep processing

---

**Algorithm 4:** Function `get_pairs` for Generating Loop Pairs

---

**Input:** List of index sequences *lst*, last node index *last_node*
**Output:** List of loop pairs *result*

1 Initialize *result* ← [];
2 **if** *lst is empty **or** all sublists in lst are empty* **then**
3     **return** *result*;
4 **foreach** *sub_lst in lst* **do**
5     **if** *sub_lst is empty* **then**
6         **continue**;
7     **if** *sub_lst is the last sublist **and** last_node* $<$ *sub_lst*[0] **then**
8         **for** $i \leftarrow 0$ **to** $|sub\_lst| - 1$ **do**
9             Append $(sub\_lst[i] - 1, sub\_lst[i])$ to *result*;
10     **else**
11         **for** $i \leftarrow 1$ **to** $|sub\_lst| - 1$ **do**
12             Append $(sub\_lst[i], sub\_lst[i-1])$ to *result*;
13         **if** $sub\_lst[-1] <$ *last_node* **then**
14             Append $(sub\_lst[-1] + 1, sub\_lst[-1])$ to *result*;

15 **return** *result*

---

## B  EXAMPLE WORKFLOW FOR CONVERTING RNA SECONDARY STRUCTURES INTO DEPENDENCY STRUCTURES

To provide a more intuitive illustration of the RNA secondary structure conversion process, we present an example depicted in Figure B1, which demonstrates the transformation through a series of steps.

**Step 1: Sequence Partitioning** involves dividing the RNA sequence into stems (blue), pseudoknots (red), and loops (black) based on its secondary structure.

**Step 2: Binary Tree Construction for Stem Sequences** entails building binary trees for each stem sequence using the `Get_Pair` function.

**Step 3: Binary Tree Construction for Pseudoknot Sequences** similarly constructs binary trees for pseudoknot sequences using the same method. In the example sequence, only one pseudoknot set is present. If multiple pseudoknots exist, separate binary trees are created for each set to accurately represent their interactions.

**Step 4: Integration of Binary Trees and Root Selection** involves integrating the multiple binary trees by selecting the last node of each tree as the root and connecting the remaining last nodes to this root node, thereby forming a unified hierarchical structure.

**Step 5: Loop Sequence Connections based on root position** establishes connections among nucleotides within loop sequences based on the position of the root node.

**Step 6: Comprehensive Structure Integration and labeling** consolidates all the constructed structures—stems, pseudoknots, and loops—and assigns appropriate labels, resulting in a comprehensive dependency structure for the given RNA sequence.

This workflow ensures that the input RNA sequence, along with its stem and pseudoknot base pairs and the list of unpaired dot indices, is systematically transformed into a dependency structure with arc connections and labels.

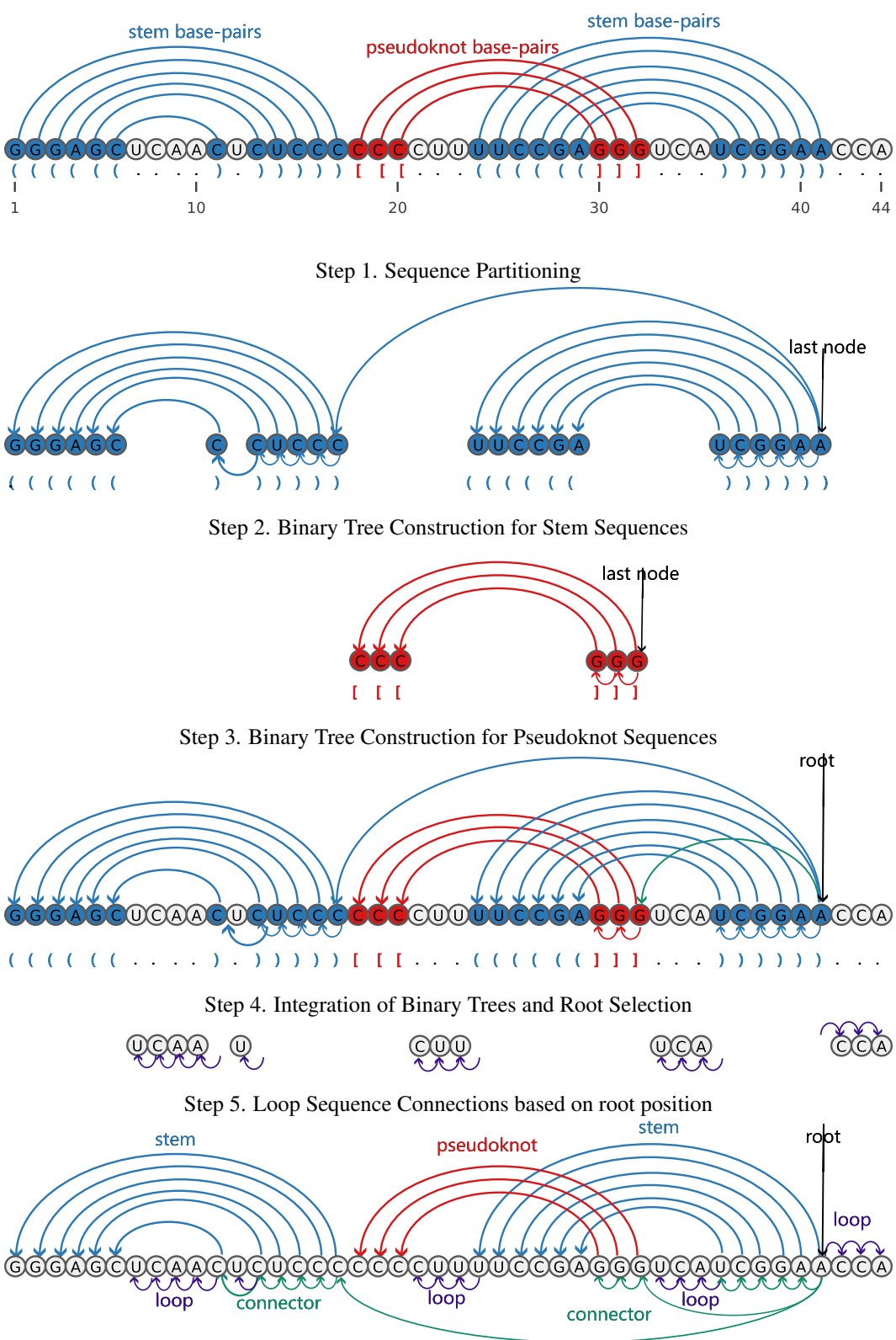

Figure B1: An example illustrating the workflow for converting RNA secondary structures into dependency structures

## C  TRAINING DETAILS

The code used in DEPfold primarily draws from parts of the SuPar (Zhang et al., 2020a;b) GitHub repository (`https://github.com/yzhangcs/parser.git`). We implemented DEPfold using PyTorch. The architecture uses `RoBERTa-base` as the encoder within a biaffine framework. Specifically, the model uses the first four layers of `RoBERTa-base`, applying mean pooling to generate a 768-dimensional representation. This encoded output is subsequently processed by two dedicated multilayer perceptrons (MLPs).

- MLP of edges: Transforms the 768-dimensional input to a 500-dimensional output.
- MLP of lables: Transforms the 768-dimensional input to a 100-dimensional output.

The outputs from these MLPs are then fed into biaffine attention layers to perform the final predictions. To mitigate overfitting, we applied a dropout rate of 0.1 to the encoder outputs and a dropout rate of 0.33 to the MLP layers.

For optimization, we used the AdamW optimizer with a dual learning rate strategy: the encoder parameters were assigned a learning rate of $5 \times 10^{-5}$, while the non-encoder parameters were set to $1 \times 10^{-3}$. This approach effectively fine-tunes the pretrained encoder while allowing the task-specific layers to adapt rapidly. The model was trained end-to-end using a cross-entropy loss function for both arc and relation predictions, with a weight decay factor of 0.01 to further prevent overfitting.

During training, we used a batch size of 32 to maximize GPU use. The training process was capped at 100 epochs, incorporating an early stopping mechanism based on the $F_1$ score on the validation set. Training was terminated when the validation $F_1$ score ceased to improve, ensuring optimal model performance and preventing overfitting. All experiments were conducted on four NVIDIA A100-40GB GPUs, enabling efficient training and scalability.

## D  INFERENCE TIME COMPARISON

We evaluated the inference time of various RNA secondary structure prediction methods on the ArchiveII dataset, as summarized in Table A1. DEPfold without Optimal Tree (OT) decoding, achieved an average inference time of approximately 0.027 seconds per sequence, markedly outperforming UFold (0.071 seconds) and LinearFold (0.075 seconds). However, the full version of DEPfold, which incorporates OT, exhibited a significant increase in inference time to 1.072 seconds per sequence, making it time-consuming among those evaluated, though still within an acceptable range.

Table A1: Inference time on the ArchiveII dataset

| Method | Time | Is pseudoknot? |
|---|---|---|
| DEPfold w/o OT (Pytorch) | 0.027s | Yes |
| DEPfold (Pytorch) | 1.072s | Yes |
| Ufold (Pytorch) | 0.071s | Yes |
| MXfold2 (Pytorch) | 0.477s | No |
| RNAfold (C) | 0.134s | No |
| ContraFold (C++) | 0.390s | No |
| Linearfold (C++) | 0.075s | No |
| RNAstructure (C) | 4.454s | Yes |

This disparity underscores the trade-off between prediction quality and computational efficiency introduced by OT, primarily attributable to the increased complexity of parsing pseudoknot structures. Despite the higher inference time, both versions of DEPfold retain the capability to handle pseudoknot structures, a feature absent in many traditional methods such as RNAfold and ContraFold. As a result, DEPfold without OT offers maximum inference speed, ideal for rapid predictions. Conversely, the full DEPfold version, while slower, delivers higher predictive accuracy by effectively modeling complex pseudoknots. This versatility makes DEPfold a robust tool for RNA secondary structure prediction, allowing users to balance speed and accuracy according to their research needs.

## E  VISUALIZATION

To intuitively demonstrate DEPfold's prediction performance, we visualized the predicted structures of three representative RNA sequences using ViennaRNA (Lorenz et al., 2011a). As illustrated in Figure E2, these sequences correspond to a short RNA, a medium-length RNA without pseudoknots, and a long RNA with pseudoknots, respectively. DEPfold's predicted results are highly consistent with the ground truth structure. In contrast, the energy-based method ContraFold failed to effectively

predict this secondary structure. Although UFold achieved a higher $F_1$ score than other baseline methods, its predicted results visually differed significantly from the ground truth and erroneously predicted pseudoknots that should not exist. DEPfold accurately captured these complex structural features.

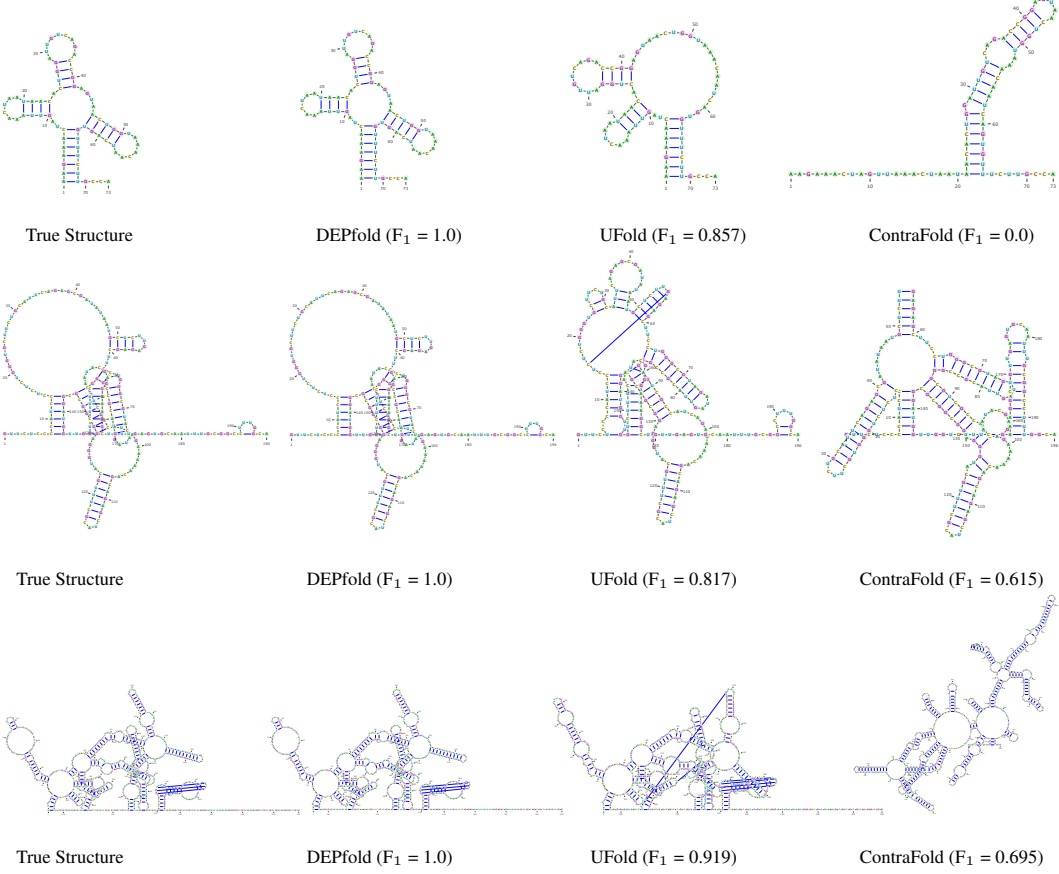

Figure E2: Comparison of RNA secondary structure predictions.

