# OpenReview forum: "DEPfold: RNA Secondary Structure Prediction as Dependency Parsing."
_ICLR.cc/2025/Conference — ICLR 2025 Poster_

### Official Review · Reviewer_vMAF · 2024-11-03

**Soundness:** 3
**Presentation:** 2
**Contribution:** 3
**Rating:** 6
**Confidence:** 4

**Summary:**

This paper introduces a novel method named DEPfold, which reframes the prediction of RNA secondary structures as a dependency parsing task.

During training, given an RNA sequence, DEPfold constructs a dependency tree from the sequence using three types of arcs/labels: stem, pseudoknot, and connector. DEPfold then employs Biaffine parsers to learn these trees.
During inference, DEPfold first predicts the tree from the raw sequence and subsequently recovers the internal secondary structures through several post-processing steps.

The authors have experimented with various base models, including RNAfm and RoBERTa, and different learning strategies such as fine-tuning and freezing pretrained models. They found that DEPfold outperforms existing models across multiple datasets in the field of RNA structure prediction.

**Strengths:**

This is a novel work, which, to my knowledge, is the first to frame RNA structure prediction as dependency parsing.

I am excited to see that the wisdom from traditional NLP parsing tasks can still play important roles in a wider range of structured prediction tasks, such as RNA secondary structures.
DEPfold is built on the biaffine parser, a widely known neural-based dependency parser, and demonstrates very strong performance by leveraging the power of pretrained masked language models like RoBERTa.

I am very optimistic about further improving the performance of DEPfold by scaling it to larger model sizes and larger datasets.
Also, beyond the scope of this work, I believe DEPfold opens the door for utilizing some other parsing methods like stack-pointer [1] networks or modeling RNA structures via context-free grammars [2].

[1] Stack-Pointer Networks for Dependency Parsing
[2] Strongly Incremental Constituency Parsing with Graph Neural Networks

**Weaknesses:**

* I suggest that the authors elaborate further on the definitions of RNA structures. For instance, a small figure illustrating the differences between stem and pseudoknot would be helpful. As someone who is not an expert in RNA structure prediction, I find it challenging to fully understand the details of the paper without resorting to external resources like Wikipedia or search engines.
* Line 284, Decoding: The Matrix-Tree Theorem is used for calculating the normalization term of the probabilities of non-projective trees. For decoding, one should use O(N^2) MST algorithms to decode non-projective trees.
* Lines 126-129: It would be beneficial if the authors could provide examples to demonstrate the bracketing structures.

**Questions:**

* > Pseudoknots, we find, are equivalent to adding a certain level of non-projectivity to the dependency parsing model.

I can't build the connection between the non-projectivity and Pseudoknots. Can the authors give me more details?

* There are some length limitatations for RoBETa that make it hard to extrapolate to longer sequences, how did you handle RNA sequence with thoundands of nucleotides?

* I'm very curious about the effectiveness of extending DEPfold to structured learning algorithms like TreeCRF, which has been proven very useful in the field of dependency Parsing [1,2,3]

* How about the speed of optimal tree decoding. Did you use any parallelized implementations of MST/Eisner in [torchstruct](https://github.com/harvardnlp/pytorch-struct) or [SuPar](https://github.com/yzhangcs/parser) for acceleration?

[1] Neural Probabilistic Model for Non-projective MST Parsing
[2] Efficient Second-Order TreeCRF for Neural Dependency Parsing
[3] Headed-Span-Based Projective Dependency Parsing

---

> ### Author Response · Authors · 2024-11-26
>
> Dear reviewer:
>
>
> Thank you for your detailed feedback. We appreciate the time you took to provide us with these valuable insights. Below, we address each of your concerns in detail:
>
>
> 1.Regarding the Definitions of RNA Structures and Examples for Bracketing Structures:
>
> Thank you for pointing this out. We have revised the description of RNA secondary structures in Section 2 to provide more precise definitions (see lines 125-141), including a more detailed explanation of pseudoknots, emphasizing their significance and structure to ensure readers have a clearer understanding. Specifically, we have redrawn Figure 1 to include a more comprehensive example, featuring stems, loops, and pseudoknots, with clear labels for each type of structure. We have also added the corresponding bracket-dot notation for these elements, making it easier to understand the relationship between the different structures and their representations.
>
>
> 2. Regarding Decoding with the Matrix-Tree Theorem:
>
> Thank you for this observation. We have corrected the wording in the paper (see line 292). In the case of inference with this algorithm, the Matrix-Tree Theorem is used to calculate the marginals over the different edges before running the maximum spanning tree algorithm.
>
>
> 3. Regarding Pseudoknots and Non-Projectivity:
>
>  The lack of pseudoknots implies that the trees generated would be projective, i.e. there would be no crossing arcs in the dependency tree. This is because the tree can be described in that case using a phrase-structure-like tree. We now provide more information about pseudoknots and nonprojectivity in section 2, and you can see that a pseudoknot is essentially a case with interlacing base pairs connected, which exactly denote a certain level of non-projectivity.
>
>
> 4. Regarding Handling Long RNA Sequences with RoBERTa:
>
> Given that RNA sequences can be very long (over 1500 tokens), we handle them by splitting the sequences into overlapping subsequences based on RoBERTa's maximum length (e.g., 512 tokens) with a suitable stride (e.g., 256 tokens) to maintain continuity of contextual information. Each subsequence is encoded independently through RoBERTa, and the outputs are then concatenated and pooled appropriately. This approach respects the model's input length limitation while preserving the complete information of long sequences, enabling RoBERTa to effectively process ultra-long RNA sequences.
>
>
> 5. Regarding Exploring TreeCRF for DEPfold:
>
> Your suggestion to extend DEPfold to structured learning algorithms like TreeCRF is insightful. We have experimented with TreeCRF in preliminary studies, but we found that training was relatively slow, possibly due to computational constraints. However, we agree that this is a promising direction for future work, as TreeCRF has demonstrated effectiveness in dependency parsing. We have added a note about this in line 530, and also referenced the stack-pointer approach, which is an excellent suggestion. We appreciate your input on this matter.
>
> 6. Regarding the Speed of Optimal Tree Decoding:
>
> Regarding the speed of optimal tree decoding, we did use parallelized implementations of MST/Eisner for acceleration. Specifically, our implementation leveraged SuPar, as noted in Appendix C, lines 1028-1029. This allowed us to achieve faster and more efficient tree decoding. We also commit to making our code publicly available to ensure transparency and reproducibility.
>
>
> Thank you again for your valuable feedback. Your comments have significantly improved the clarity and depth of our paper. We hope our revisions and explanations adequately address your concerns.

---

> > ### Comment · Reviewer_vMAF · 2024-11-26
> >
> > Thank you for your hard work and clarifications, which addressed most of my concerns.
> > In light of the comments from other reviewers, I maintain my positive score.

---

> > > ### Author Response · Authors · 2024-11-28
> > >
> > > Dear reviewer:
> > >
> > > Thank you very much for your positive comments!
> > >
> > > Best regards,
> > >
> > > The Authors

---

### Official Review · Reviewer_SsoD · 2024-11-04

**Soundness:** 3
**Presentation:** 3
**Contribution:** 3
**Rating:** 6
**Confidence:** 2

**Summary:**

The paper proposes to use deep dependency parsing technique to predict RNA secondary structures. The paper shows how RNA structures can be mapped to dependency structures and via verse. Then now, the problem of predicting RNA structures is casted to dependency parsing, a classical task in NLP. The paper uses the Dozat & Manning 2016 parsing method, in which nucleotides are represented by RNA-fm or Roberta contextual embeddings.

The paper shows experiments with popular datasets, RNAStrAlign, ArchiveII, and bpRNA-*. The proposed method substantially outperforms strong baselines (UFold, MXfold2, E2Efold). The paper also presents analyses showing a surprising result that Roberta contexutal embeddings actually are effective for this problem.

**Strengths:**

This paper showcases a very interesting application of NLP (more specifically, dependency parsing) to biology. The authors propose an effective way to translate the RNA structure prediction to dependency parsing, connecting the two challenges. The experiment results are evidence for the effectiveness of the proposed method.

**Weaknesses:**

The paper only mentions methods and work from 2022 backward, and only one published in 2024. However, I found several publications in 2023, such as [1,2,3]. I thus question about the up-to-date information in the paper.

(I'm willing to raise my overall score if the authors can provide comparison with the most up-to-date work in the literature).


1. Chen, CC., Chan, YM. REDfold: accurate RNA secondary structure prediction using residual encoder-decoder network. BMC Bioinformatics 24, 122 (2023). https://doi.org/10.1186/s12859-023-05238-8
2. Wang, W., Feng, C., Han, R. et al. trRosettaRNA: automated prediction of RNA 3D structure with transformer network. Nat Commun 14, 7266 (2023). https://doi.org/10.1038/s41467-023-42528-4
3. Tzu-Hsien Yang. DEBFold: Computational Identification of RNA Secondary Structures for Sequences across Structural Families Using Deep Learning. Journal of Chemical Information and Modeling 2024 64 (9), 3756-3766. DOI: 10.1021/acs.jcim.4c00458

**Questions:**

* Did the authors try "undirected" dependency structures? if yes, what are the performance?
* Because a RNA sequence can be very long (>1500), could the authors explain how to use Roberta effectively?

---

> ### Author Response · Authors · 2024-11-26
>
> Dear Reviewer,
>
> Thank you for your valuable feedback and for bringing these recent publications to our attention.
>
> 1. Regarding the lack of up-to-date comparisons, with recent publications from 2023 and 2024:
>
> Thank you for highlighting this point. Indeed, several new deep learning methods have emerged recently. However, due to the limitations and inconsistencies of available RNA datasets, we aimed to provide a comprehensive and fair evaluation of the generalization capabilities of our model using well-established benchmarks.
>
> Specifically, we have reviewed the three works you mentioned:
>
> REDfold: This work does not provide source code for training but only a packaged prediction program. Its training data includes an Rfam dataset that differs from our training datasets, making a direct comparison challenging. The framework used by REDfold is based on a U-net architecture similar to that of UFold, which we have already included in our comparisons. Therefore, we believe our comparison with UFold is representative, especially given that our approach differs fundamentally from these methods.
>
> trRosettaRNA: This model is designed for RNA 3D structure prediction, which is a different task compared to our work focusing on RNA secondary structure prediction. Therefore, a direct comparison is not feasible.
>
> RNAformer: This work only provides inference code with pre-trained parameters, and it is not possible to retrain it on the same datasets used in our study for a fair comparison. The model is trained on a mixture of databases, whereas we train on one dataset at a time to evaluate specific capabilities of our model. In our attempt to test RNAformer on the bpRNA-TS0 dataset, it achieved an F1 score of 0.706, which is quite high. However, when tested on new families from the bpRNA-new dataset, its F1 score dropped significantly to 0.394, suggesting potential overfitting issues.
>
>
> In response to your suggestion, we have added a comparison with a newer model, RFold[1], published at ICML 2024, which is a state-of-the-art model in this domain. Our model demonstrates superior performance compared to RFold under the same training and testing conditions. These results have been added to the revised paper, as shown in Tables 2-8.
>
> We commit to making our source code and model weights publicly available, along with all comparison data, to ensure full transparency and reproducibility.
>
> 2. Regarding undirected dependency structures:
>
> We have only utilized directed dependency structures in our work. Thank you for this suggestion; we will consider exploring undirected structures as a direction for future research.
>
> 3. Regarding handling long RNA sequences effectively with RoBERTa:
>
> Given that RNA sequences can be very long (over 1500 tokens), we handle them by splitting the sequences into overlapping subsequences based on RoBERTa's maximum length (e.g., 512 tokens) with a suitable stride (e.g., 256 tokens) to maintain continuity of contextual information. Each subsequence is encoded independently through RoBERTa, and the outputs are then concatenated and pooled appropriately. This approach respects the model's input length limitation while preserving the complete information of long sequences, enabling RoBERTa to effectively process ultra-long RNA sequences.
>
>
> Thank you again for your constructive comments, which have significantly helped us improve our paper. We hope that our revisions and explanations address your concerns and demonstrate the up-to-date nature and robustness of our work.
>
> [1] Cheng Tan, Zhangyang Gao, CAO Hanqun, Xingran Chen, Ge Wang, Lirong Wu, Jun Xia, Jiangbin Zheng, and Stan Z Li. Deciphering rna secondary structure prediction: A probabilistic k-rook matching perspective. In Forty-first International Conference on Machine Learning.

---

> ### Author Response · Authors · 2024-11-28
>
> Dear Reviewer,
>
> As the deadline for the rebuttal phase approaches, we kindly ask you to confirm whether we have sufficiently addressed your comments or if there are any remaining concerns. More specifically, you mentioned you would be willing to increase your score if we follow up on more recent work, and we included new results (RFold) as mentioned in our rebuttal. We discussed other issues you mentioned there.
>
> Thank you very much for your feedback!
>
> Best regards,
>
> The Authors

---

> > ### Comment · Reviewer_SsoD · 2024-12-02
> > **response**
> >
> > I would like to thank the authors for the satisfying answer. I thus raised the score to a positive score.

---

### Official Review · Reviewer_XEMD · 2024-11-04

**Soundness:** 2
**Presentation:** 3
**Contribution:** 3
**Rating:** 6
**Confidence:** 3

**Summary:**

This paper introduces a new NLP dependency parsing-inspired algorithm for RNA secondary structure prediction, DEPfold. The paper's results suggest that DEPfold performs much better than other methods on within-family and cross-family RNA datasets, getting near perfect results on several datasets.

As a "positionality statement" for this review, I'll mention that I'm expert in NLP dependency parsing, and assume that I was chosen to review for that reason, but I have little understanding of the biology and am generally not familiar with the test sets and performance of other methods in the RNA structure prediction domain.

**Strengths:**

- The approach to treating RNA secondary structure prediction via NLP dependency parsing, and largely using the biaffine dependency parsing algorithm of Dozat and Manning (2016) seems largely original.
- The results of the paper, as presented, are very strong. If this all checks out, the method provides a new much more accurate RNA secondary structure prediction and this would be quite significant

**Weaknesses:**

- I feel that the presentation of the paper should have been clearer. It might have been better to move Fig. 1 to section 3 and to have referred to it when presenting the algorithm. The algorithm for conversion to dependency trees is presented somewhat informally. Something more precise would have been better. Among other things: (i) The algorithm is presented with reference to "bracket-dot notation" for RNA secondary structure (lines 126-136) but this is never rigorously defined. This notation may be well known in bioinformatics, but not to me. If this is the starting point of the algorithm, it would minimally be really useful to have shown the example sequence in Fig. 1 in bracket-dot notation. Indeed, it seems like the two subdiagrams in the top right of Figure 1 do not add much to understanding and at least one of them could have been deleted. Lines 165-187: While a projective dependency diagram is isomorphic to a certain kind of binary tree (a single-level X' CFG), a pure dependency grammar presentation would not normally evoke tree structure. Is it necessary? I would guess not and that the algorithm could be described directly in terms of creating dependency arcs. Things might even be clearer that way, given what is in Fig. 1. Tree completion (lines 210-215): AFAICS, g is never defined. Is it the root node, or the "grandparent" (parent of the head node)? Most importantly, I don't think the section 3.1 presentation of the algorithm even attempts to explain rigorously the treatment of pseudoknots. They're mentioned, as on line 223, but what is the details of their dependency grammar representation. Are they just treated as some kind of not fully labeled clump, as perhaps suggested by the notation introduced for P_i on line 159? I know vaguely that there is an older thread of work crossing from NLP to bioinformatics arguing that giving powerful enough grammars to describe pseudoknots requires more powerful "mildly context sensitive" grammars like tree-adjoining grammar, beyond the power of dependency grammars (e.g., https://academic.oup.com/bioinformatics/article/21/11/2611/294713 ). Is that still true? Is having a non-projective dependency tree grammar sufficient? This paper left me no wiser. Finally, I wondered whether the post-processing (line 291ff) couldn't have been incorporated into the decoding algorithm with some appropriate constraint.

 - Results: I'm not really the person to judge the appropriateness and completeness of all the results presented, but to the extent that I tried to look other things up on the web for a few minutes, I seemed to be left with more questions than answers. The algorithm referred to in this paper as "E2Efold" is referred to by the authors of the cited reference as E2Efold-3D. There is actually a different algorithm by different authors called E2Efold that appear at ICLR in 2020 (https://openreview.net/forum?id=S1eALyrYDH) and which itself used an algorithm "close to" biaffine dependency parsing. At least it is mentioned as related work. How does this paper refer to that paper. This paper is not in the references, and there is no explicit comparison in related work. But then the RNAStrAlign results in Table 2 seem to be the results of E2Efold not E2Efold-3D. They're identical to the ones in the E2Efold paper. Somehow this isn't giving me confidence.... I don't really know what are the best datasets or perceived best methods in this domain, but it then also seems like there are other recent methods claiming good results, such as RNAformer (https://icml-compbio.github.io/2023/papers/WCBICML2023_paper43.pdf) or DEBFold (https://pubs.acs.org/doi/10.1021/acs.jcim.4c00458) which aren't mentioned in the references here. Am I getting the best most up-to-date comparisons? It's not clear to me. The latest, best algorithm compared to is UFold from 2022, but there are clearly papers on this topic from 2023 and 2024, but it's not trivial for me to compare since there seem to be a lot of different datasets around, etc.

**Questions:**

This largely duplicates what I put in "weaknesses".

- Lines 165-187: While a projective dependency diagram is isomorphic to a certain kind of binary tree (a single-level X' CFG), a pure dependency grammar presentation would not normally evoke tree structure. Is it necessary? I would guess not and that the algorithm could be described directly in terms of creating dependency arcs. Things might even be clearer that way, given what is in Fig. 1.

- Tree completion (lines 210-215): AFAICS, g is never defined. Is it the root node, or the "grandparent" (parent of the head node)?

- Most importantly, I don't think the section 3.1 presentation of the algorithm even attempts to explain rigorously the treatment of pseudoknots. They're mentioned, as on line 223, but what is the details of their dependency grammar representation. Are they just treated as some kind of not fully labeled clump, as perhaps suggested by the notation introduced for P_i on line 159? I know vaguely that there is an older thread of work crossing from NLP to bioinformatics arguing that giving powerful enough grammars to describe pseudoknots requires more powerful "mildly context sensitive" grammars like tree-adjoining grammar, beyond the power of dependency grammars (e.g., https://academic.oup.com/bioinformatics/article/21/11/2611/294713 ). Is that still true? Is having a non-projective dependency tree grammar sufficient?

- Can the post-processing (line 291ff) be incorporated into the decoding algorithm with some appropriate constraint?

- Clarify the relation between this algorithm and E2Efold (from ICLR 2020)

- Clarify the relation and what you're citing as results between E2Efold and E2Efold-3D

- Clarify whether there are results from 2023 or 2024 that should be included as comparisons and how the methods and results of algorithms from these years relate to yours.

---

> ### Author Response · Authors · 2024-11-26
>
> Dear reviewer:
>
> Thank you for your detailed feedback. We appreciate the time and effort you took to review our work and provide such insightful comments. Below, we address each of your concerns in detail:
>
> 1. Regarding Figure1 and Bracket-Dot Notation:
>
> Thank you for pointing this out. We have revised the description of RNA secondary structures in Section 2 to provide more precise definitions (see lines 125-141), including a more detailed explanation of pseudoknots, emphasizing their significance and structure to ensure readers have a clearer understanding. Specifically, we have redrawn Figure 1 to include a more comprehensive example, featuring stems, loops, and pseudoknots, with clear labels for each type of structure. We have also added the corresponding bracket-dot notation for these elements, making it easier to understand the relationship between the different structures and their representations. As you suggested, we have appropriately cite the Figure 1 in Section 3.
>
> 2. Regarding the  Presentation of the Algorithm for Conversion to Dependency Trees:
>
> To address this, we have added pseudocode in Appendix A that outlines each step of the transformation in detail. Moreover, we have provided a step-by-step example in Appendix B that visually demonstrates how the RNA structure is transformed into a dependency tree, making the process more intuitive for readers.
>
> 3. Regarding the Necessity of Tree Representation:
>
> The tree constraint enforces that much of the structure is projective overall, which is indeed the case for RNA structures, where pseudoknots are rarer (and denote non-projectivity). As such, the use of a tree adds further structural constraints that improve the inductive bias of the algorithm. Indeed, RNA structure prediction has been done in the past using CFGs, which denote similar tree nestedness constraints.
>
> 4. Clarification of Node $g$ in Tree Completion:
>
> Thank you for pointing this out. Node $g$ refers to the node which connect to root node of the entire sequence's dependency tree. We have now clarified this definition in line 119 and revised the description in lines 119-219. Additionally, we have included pseudocode in Appendix A and an example in Appendix B to facilitate understanding.
>
> 5. Regarding the Treatment of Pseudoknots in the Algorith:
>
> In our algorithm, pseudoknots and stems are treated as two independent sequences first, and each generating its own tree structure using the same method before being connected together. Please refer to the detailed example provided in Appendix B for a step-by-step illustration.
>
> 6. Regarding the Power of Grammars for Describing Pseudoknots:
>
> Thank you for the valuable reference! We have added it (see line 493). We now include more information about pseudoknots in Section 3, explaining that a pseudoknot is essentially a case with interlacing base pairs, which denotes a certain level of non-projectivity. Thus, there is a clear mapping between non-projectivity and pseudoknots.
>
> 7. Regarding Post-Processing and Decoding Algorithm:
>
> Indeed, we are currently unable to incorporate these constraints directly into the decoding algorithm, but this is a promising idea and something we will explore in future work. We appreciate your suggestion.
>
> 8. Clarification on the Relation to E2Efold-3D:
>
> We sincerely appreciate your close reading and for pointing out the confusion regarding the reference to E2Efold. Indeed, this was an oversight on our part. We intended to cite the original E2Efold paper, not E2Efold-3D. We have now corrected the reference throughout the paper to accurately cite E2Efold and have checked all other citations to prevent similar mistakes. See line 571.

---

> ### Author Response · Authors · 2024-11-26
>
> 9. Regarding Comparisons with Results from 2023 or 2024:
>
> Regarding recent methods, we acknowledge that there have been new developments in the field in the last couple of years. The limitations of RNA datasets and inconsistencies among them make it challenging to establish a comprehensive and fair comparison for all recent models. However, based on your suggestion, we have reviewed the algorithms mentioned.
>
> RNAformer: This work only provides inference code with pre-trained parameters, and it is not possible to retrain it on the same datasets used in our study for a fair comparison. The model is trained on a mixture of databases, whereas we train on one dataset at a time to evaluate specific capabilities of our model. In our attempt to test RNAformer on the bpRNA-TS0 dataset, it achieved an F1 score of 0.706, which is quite high. However, when tested on new families from the bpRNA-new dataset, its F1 score dropped significantly to 0.394, suggesting potential overfitting issues.
>
> DEBFold: We noted that only a web-based prediction interface is available, and the source code is not provided, making retraining on our datasets impossible. Moreover, DEBFold is trained on the Rfam dataset, which is inconsistent with our datasets, making a direct comparison difficult. Given these limitations, it is not feasible to obtain a fair comparison with our model.
>
> In response to your suggestion, we have added a comparison with a newer model, RFold[1], published at ICML 2024, which is a state-of-the-art model in this domain. Our model demonstrates superior performance compared to RFold under the same training and testing conditions. These results have been added to the revised paper, as shown in Tables 2-8.
>
> We hope these revisions and additional explanations address your concerns and help resolve them effectively. Thank you again for your invaluable feedback.
>
>
> [1] Cheng Tan, Zhangyang Gao, CAO Hanqun, Xingran Chen, Ge Wang, Lirong Wu, Jun Xia, Jiangbin Zheng, and Stan Z Li. Deciphering rna secondary structure prediction: A probabilistic k-rook matching perspective. In Forty-first International Conference on Machine Learning.

---

> ### Comment · Reviewer_XEMD · 2024-11-28
> **Definitely improved**
>
> Thank you for your responses to all my questions.
>
> The paper is definitely improved.
>
> I'm still a bit unsure about some of the issues of comparison of different methods and the formal language class that is needed to handle pseudo-knots. I'm afraid that I'm only looking at the new version sort of quickly.
>
> I'm definitely raising my score to a 6. It could be that this paper is worth more?

---

> > ### Author Response · Authors · 2024-11-28
> >
> > Dear reviewer:
> >
> > Thank you very much for your positive comments!
> >
> > Best regards,
> >
> > The Authors

---

### Official Review · Reviewer_ubS6 · 2024-11-04

**Soundness:** 3
**Presentation:** 3
**Contribution:** 3
**Rating:** 6
**Confidence:** 3

**Summary:**

After the author response, I appreciate the efforts that the authors made to make the paper more readable. I especially found the more explicit inclusion of pseudoknots and the workflow in Figure B1 quite helpful. I thus adjusted my score from a 5 to a 6.

----

The paper presents a novel and interesting approach for RNA secondary structure prediction. The authors frame this problem as dependency parsing in NLP. This first involves mapping the RNA structure to a dependency tree (this can be challenging because of pseudo-knots that result in non-projectivity).

The authors then use a neural network biaffine parser (based on Dozat and Manning - 2016) for this task which predicts the pairwise edges + labels. The authors then use either Eisener algorithm (for projective structures or Kirchoff's Matrix-Tree Theorem (for non-projective structure) to find the optimal tree.

Experiments show that the authors achieve strong performance on multiple datasets.

**Strengths:**

-The paper tackles an important problem.

-The method is interesting and novel.

-Empirical results are strong.

**Weaknesses:**

Overall I found the description of the RNA secondary structure and how to transform it into a dependency tree to be confusing and non-rigorous.

-There is brief notation given in Section 2 and it states how a typical RNA secondary structure is composed of linear sequences (dots) and then different types of brackets. However, this is all quite vague and non-rigorous.

 It would be great to see an example of each type of label and how it is represented in this notation (e.g. stems, loops, pseudoknots etc.) I feel these concepts are not rigorously defined i.e. despite the importance of pseudo-knots I do not see a clear example/definition in the paper.

-The description of 3.1 (transformation of RNA structures to dependency) would be clearer with a running example that shows what happens at each step. Moreover, I am confused about whether the transformation is a 1:1 mapping. Is it provable the case that for any RNA secondary structure it maps to a unique dependency tree and vice versa? Or is this a heuristic.

**Questions:**

I have many questions about the RNA secondary structure and the transformation to dependency trees as described above in Weaknesses.

---

> ### Author Response · Authors · 2024-11-26
>
> Dear Reviewer,
>
> Thank you for your valuable feedback. Based on your suggestions, we have made several adjustments to improve the clarity and rigor of our paper.
>
> 1. Regarding the vague and non-rigorous description of RNA secondary structures in Section 2, including the need for examples and clearer definitions of pseudoknots:
>
> Thank you for pointing this out. We have revised the description of RNA secondary structures in Section 2 to provide more precise definitions (see lines 125-141), including a more detailed explanation of pseudoknots, emphasizing their significance and structure to ensure readers have a clearer understanding. Specifically, we have redrawn Figure 1 to include a more comprehensive example, featuring stems, loops, and pseudoknots, with clear labels for each type of structure. We have also added the corresponding bracket-dot notation for these elements, making it easier to understand the relationship between the different structures and their representations. This should provide a clearer and more thorough introduction for readers without extensive background knowledge.
>
> 2. Regarding the transformation of RNA structures to dependency trees in Section 3.1:
>
> We agree that including a running example would significantly improve the clarity of the transformation process. To address this, we have added pseudocode in Appendix A that outlines each step of the transformation in detail. Moreover, we have provided a step-by-step example in the appendix B that visually demonstrates how the RNA structure is transformed into a dependency tree, making the process more intuitive for readers.
>
> 3. Regarding the uniqueness of the mapping between RNA secondary structures and dependency trees:
>
> Thank you for raising this important question. In our approach, we use a right-to-left tree generation and connection method, along with specific constraints that ensure a consistent mapping. Under these directional and rule-based constraints, the transformation from an RNA secondary structure to a dependency structure is indeed unique. Similarly, the reverse transformation from the dependency structure back to the RNA secondary structure is also unique. As shown in Figure 1 and Appendix B, this consistency is clearly demonstrated. Of course, these constraints are not the only possible approach—alternative directions, such as left-to-right, or other connection methods could also be employed.
>
> We sincerely appreciate your constructive feedback, which has greatly helped us improve the clarity and rigor of our paper. We hope that our revisions and additional explanations effectively address your concerns.

---

### Meta-Review · Area_Chair_JdMk · 2025-01-03

**Metareview:**

The authors propose a new method for predicting RNA secondary structure by reframing the task as a dependency parsing problem. Their approach, called DEPfold, involves three main steps: generating a dependency tree from the RNA structure, using an attention-based framework to predict the tree’s elements, and then decoding this predicted tree into RNA sequences under structural constraints. Tests show that DEPfold outperforms existing methods and generalizes well across both in-family and cross-family.

During the review process, several reviewers found the initial description of the method unclear. In response, the authors significantly revised Section 3 and added a pseudocode example with a working illustration in Appendix A. These updates satisfied reviewers ubS6 and XEMD, who raised their scores to 6. Reviewers also noted that recent methods (from 2022 to 2024) were missing from the comparisons. To address this, the authors explained why techniques like RNAFormer and DEBFold were not directly comparable, and they incorporated Rfold as a baseline in Tables 2–8 to benchmark DEPfold against newer work.

The authors also improved the paper’s readability, including improving the explanation of bracket-dot notation (lines 125–141), refining the definition of node g (line 119), and clarifying the discussion of pseudoknots in Section 3. Following these changes, all reviewers agreed that the paper should be accepted.

**Additional Comments On Reviewer Discussion:**

See above.

---

### Decision · Program_Chairs · 2025-01-22

Accept (Poster)